# Estimating the prevalence and characteristics of people in severe social isolation in 29 European countries: A secondary analysis of data from the European Social Survey round 9 (2018–2020)

**Simone Amendola**[1]*, **Rita Cerutti**[2], **Agnes von Wyl**[1]

1 Department of Applied Psychology, Zurich University of Applied Sciences, Zurich, Switzerland,
2 Department of Dynamic and Clinical Psychology, Health Studies, Sapienza–University of Rome, Rome, Italy

* amen@zhaw.ch

**Data Availability Statement:** The data underlying the results presented in the study are available at https://ess-search.nsd.no/.

## Abstract

The main aim of the present study was to estimate the prevalence of people in severe social isolation as a proxy for high risk of hikikomori using data from 29 European countries. The relationship between the presence/absence of severe social isolation and demographic and psychosocial variables was also investigated. Publicly available data from the European Social Survey (ESS) round 9 collected between August 2018 and January 2020 were used. Data from the ESS round 1 (September 2002 –December 2003) and round 10 (September 2020 –May 2022) were also examined to investigate changes in the prevalence of severe social isolation over time. Analyses were restricted to the working-age population (15–64 years). A complex sampling design to obtain weighted prevalence and results was used. The study protocol was preregistered online on the Open Science Framework (https://osf.io/6a7br/). The weighted prevalence of severe social isolation was 2.01% for the sample from the ESS 1, 1.77% for the sample from the ESS 9, and 1.71% for the sample from the ESS 10, indicating a decrease over time, mainly in males. Logistic regression models showed that different sociodemographic factors (e.g., being retired, being permanently sick or disabled, doing housework, living in Central and Eastern Europe, living uncomfortably on household income, having no income) were associated with severe social isolation. Further, feeling unsafe when walking alone in the neighbourhood after dark, low social trust, and support, decreased happiness and lack of future planning correlated with severe social isolation after adjustment for the effect of sociodemographic factors was made. In this study, the prevalence of severe social isolation as a proxy for hikikomori in European countries is in line with that found by previous representative studies conducted in Asian countries. The novelty of the findings as well as implications for hikikomori research are discussed according to recent scientific literature.

**Funding:** The authors received no specific funding for this work.

**Competing interests:** The authors have declared that no competing interests exist.

# Introduction

During the last decade, the number of scientific studies on social withdrawal and social isolation has constantly increased attesting to the substantial attention from researchers and public health experts. Although both terms are sometimes used interchangeably in the literature, they have different meanings. Social withdrawal refers to the avoidance of or disinterest in social interactions with others [1] and together with peer/active isolation may explain the lack of social interaction [1,2]. Social withdrawal has been studied less than social isolation, mainly in children [3], and as a symptom of some mental health disorders [4]. While the concept of social isolation indicates "both objective social contact and subjective perceived adequacy of contact" [5] (p. 1453), that is, inadequate quality and quantity of social relations. In accordance with the authors [5], social isolation contains a subjective judgement besides a more objective one while loneliness is entirely subjective. It has been mainly studied with samples of older adults and in relation to mental health due to its impact on mortality and morbidity [6–8].

## Hikikomori

*Hikikomori* is a specific and extreme form of social isolation and withdrawal that has attracted scientific attention worldwide. According to the most recent definition, "hikikomori is a form of pathological social withdrawal or social isolation whose essential feature is physical isolation in one's home" [9] (p. 431). Marked social isolation at home, withdrawal duration of at least six months, and significant impairment or distress associated with the social isolation are core criteria indicating the presence of hikikomori [9]. Specifically, individuals who leave their homes a maximum of two-three days per week may be categorized as hikikomori, instead, those who leave their homes four or more days per week would not be considered in hikikomori [9].

There have been several attempts over time to define the hikikomori condition [10]. Specifically, avoidance, disinterest or unwillingness to attend school/work and to participate in social relationships/interactions, as well as the absence of other primary mental disorders have been considered core criteria of hikikomori [11–13]. In addition, the presence of a chronic physical illness and an accident that justifies social withdrawal have been proposed as exclusion criteria for hikikomori [11]. Therefore, the recent definition of hikikomori by Kato et al. [9] is broader compared to previous ones. Importantly, despite the authors defined hikikomori as "a form of pathological social withdrawal" (p. 431), avoidance of social participation (e.g., attending school, work) and interaction (e.g., friendships, contact with family members) is not an essential criterion of the condition. Thus, according to the above definition, hikikomori refers to a pathological or severe form of (home-based) physical isolation.

However, a recent systematic review of the literature [14] demonstrated that more than 80% of the examined studies included the following indicators of hikikomori: not working or attending school, not socializing outside one's home, and duration of hikikomori (generally, longer than six months). Despite that, differences in the operationalization of hikikomori are frequent. For example, Yong et al. [15] operationalized the presence of hikikomori as "not having participated in any social events nor interacted with others besides family members for more than six months" (p. 237) without investigating marked isolation at home. Uchida and Norasakkunkit [16] considered hikikomori those participants who, at the time, were not in a job or seeking a job, and did not go out of their home, with few exceptions, for more than six months, without investigating social interaction. Further, these two studies as well as forty-six (88.5%) out of fifty-two studies analysed in the systematic review mentioned above [14] did not consider significant impairment or distress associated with marked isolation as an inclusion criterion for the definition of hikikomori. Therefore, further evidence is needed to analyse

how the use of different criteria adequately represents the phenomenological presentation of hikikomori influencing its interpretation [14].

Initially considered a Japanese culture-bound syndrome, cases of hikikomori have been reported worldwide. This condition has been mainly studied in Japan and other Asian countries despite initial studies have been published in different western countries [17–25].

In particular, studies conducted with representative samples showed a prevalence of hikikomori of 1.2% [26] and 1.9% [11] among adults aged 20–49 in Japan. A prevalence of 1.1% was reported among young adults aged 20–39 years [16]. Similarly, a prevalence of 1.6% was found among Japanese adolescents and adults aged 15–39 years [27]. Recently, Yong et al. [15] demonstrated a higher prevalence (6.7%), among a working-age population (15–64 years) sample, compared to previous findings in Japan. Further, two studies conducted in China showed a prevalence of 1.9% [28] and 3.2% [29] among individuals aged 12–29 and university students, respectively.

On the contrary, to the best of our knowledge, only three studies [18,19,30] conducted in western countries with convenience samples ($150 < N < 400$) of the general population provided information on frequency of hikikomori episodes, ranging from 1.1% (Italian adults) to 20.9% (Nigerian university students). Therefore, it is essential to estimate the frequency of the condition using more representative samples as well as a common definition of hikikomori across studies.

Finally, it is noteworthy that a previous study [16] demonstrated that more than half (114 of 200) of participants Not in Employment Education or Training (NEET) were in a hikikomori condition. Within the NEET group, inactive individuals (i.e., those who have not looked for work and/or are not available to work) may be at high risk compared to the unemployed (i.e., those who are looking for work but cannot find it).

## Study aim

In light of the above, the aim of the present study was to estimate the prevalence of people in severe social isolation as a proxy for high risk of hikikomori [16], using data from 29 European countries. To our knowledge, no research to date has estimated the prevalence of people in severe social isolation in the European working age population (aged 15–64). Further, the relationships between the presence/absence of severe social isolation and demographic and psychosocial variables potentially associated with the condition were investigated.

## Materials and methods

### Data

Our analysis is based on publicly available data from the European Social Survey (ESS) round 9 [31]. The survey was conducted in 29 European countries using strict random probability sampling, a minimum target response rate of 70% and rigorous translation protocols [32]. Face-to-face interviews included questions exploring a variety of social indicators. The ESS round 9 was conducted with persons aged 15 and over, resident within private households, between August 2018 and January 2020. For this study, the working-age population (aged 15–64) was considered, resulting in $N = 35,749$ (after excluding 13,771 participants not in the working age).

The ESS round 9 was preferred to the ESS round 10 (September 2020 –May 2022) because data from round 10 was only partially available (for 25 countries of a total of 40 participating countries) when data was downloaded from https://ess-search.nsd.no/ on December 22, 2022. In addition, we believe that the spread of COVID-19 and national restrictive measures might have influenced the results of the ESS 10 due to their impact on individuals' behaviour and

perception (e.g., on social meeting and social activity) potentially biasing group membership (i.e., severe social isolation vs. general population).

For comparative purposes, prevalence rates of severe social isolation using data from the ESS round 1 (September 2002 –December 2003) (edition 6.6, *N* working-age population = 33,834) and ESS round 10 (edition 2.2, *N* working-age population = 24,405) were also examined to investigate change over time.

The study protocol was preregistered online on the Open Science Framework (https://osf. io/6a7br/).

## Measures

**Severe social isolation as a proxy for high risk of hikikomori.** For the purpose of the present study, the following indicators were considered to identify individuals in severe social isolation with high risk of hikikomori based on lack of social interaction and participation: 1) social meeting with friends, relatives or colleagues less than once a month or never, 2) taking part in social activities less than most or much less than most compared to others of same age, 3) not working (or not away temporarily) during the last week, 4) not actively looking for a job during the last week and 5) not being in education (not paid for by employer), even if on vacation, during the last week. If all five indicators were present, an individual was classified as in severe social isolation. Therefore, the condition examined differs from social isolation in general, as evaluated considering objective indicators of frequency and/or number of social contacts. The inclusion of lack of social participation (not working and not searching for work, not being in education) and social relationships as defined above—but not including significant impairment and/or distress—[13,14] enabled us to examine severe social isolation as a proxy for high risk of hikikomori. All indicators are reported in S1 Table.

**Sociodemographic characteristics.** The sociodemographic characteristics explored included: sex, age, number of people living regularly as members of the household, living alone, living with husband/wife/partner, living with parent/parent-in-law/partner's parent/ step parent, ever given birth to/ fathered a child, living area (e.g., big city, small city, village), country, region of Europe according to EuroVoc divided in Northern (Denmark, Estonia, Finland, Iceland, Latvia, Lithuania, Norway, Sweden), Southern (Cyprus, Italy, Portugal, Spain), Western (Austria, Belgium, France, Germany, Ireland, Netherlands, Switzerland, United Kingdom), and Central and Eastern Europe (Bulgaria, Czech Republic, Croatia, Hungary, Montenegro, Poland, Serbia, Slovakia, Slovenia), highest level of education, years of education completed, being permanently sick or disabled, being hampered in daily activities by illness/ disability/infirmity/mental health problem, housework/looking after children/others during the last seven days, being retired, Internet use, born in country other than that of residence/ domicile, father or mother born in country other than that of residence/domicile (coded as 1 if at least one parent was born outside country), parents' highest level of education, father's employment status when respondent was 14, mother's employment status when respondent was 14, main source of household income, household's total net income (all sources), feeling about household's income nowadays and respondent's main source of income.

**Adverse events and fear.** Stressful or humiliating experiences may exert a prominent role in the onset of hikikomori [28,33]. Hence, two items were used to explore respondent or household member victimization due to burglary/assault during the last five years and feeling of safety in walking alone in local area after dark.

**Emotional support and trust.** Emotional support was investigated by asking respondents the number of people with whom he/she can discuss intimate and personal matters.

Social trust was measured using the Social Trust Scale [34]. Three items explore respondent's trust in other people, his/her belief that most people would try to take advantage of him/her, and the belief that most of the time people try to be helpful or that they are mostly looking out for themselves. Each item is rated on a 11-point Likert scale, with higher scores indicating higher social trust. A total mean score is computed summing item responses and dividing by the number of items answered.

Political trust [35] was measured using five items exploring trust in country's parliament, in the legal system, in the police, in politicians and in political parties. Each item is rated on a 11-point Likert scale, with higher scores indicating higher trust. A total mean score is computed summing item responses and dividing by the number of items answered.

**Well-being.**   Respondents' well-being was explored considering satisfaction with life, happiness and subjective general health. The attitude of planning for the future was also analysed.

## Statistical analysis

1.78% (n = 602) of the working age population of the sample from the ESS round 1 (*N* = 33,834) was excluded from the analysis due to missing data on at least one indicator considered to identify persons in severe social isolation resulting in a sample of 33,232 individuals. For the same reason, 1.61% (n = 577) of the sample from the ESS round 9 and 1.63% (n = 397) of the sample from the ESS round 10 were excluded from the analysis resulting in final samples of 35,171 and 24,008, respectively.

We used a complex sampling design to obtain weighted prevalence and results. The analysis weight variable is suitable for all analyses, such as comparing multiple countries or studying multiple countries as a group [36].

Descriptive statistics (means, standard deviations, prevalence) were used to explore the characteristics of the following two groups of participants 1) no severe social isolation (i.e., general population) and 2) severe social isolation with high risk of hikikomori. Confidence intervals (non-) overlap was used for identifying significant differences in prevalence of severe social isolation across the ESS rounds 1, 9 and 10.

Between groups differences were tested using Chi-square tests of independence and t-tests for categorical and continuous dependent variables, respectively.

Finally, logistic regression models were fitted to test adjusted associations between presence/absence of severe social isolation (dependent variable) and the following variables: a) sociodemographic characteristics, b) adverse events and fear, c) emotional support and trust and d) well-being indicators adjusting for sociodemographic characteristics.

All analysis were performed in RStudio using the "survey" package [37].

## Results

### Prevalence and trend of severe social isolation

Weighted prevalence of severe social isolation was 2.01% for the ESS 1 sample, 1.77% for the ESS 9 sample, and 1.71% for the ESS 10 sample (Table 1). The decrease in total prevalence over time was not significant (i.e., confidence intervals overlap). Conversely, the decrease became significant when only ten countries with available data in each ESS round were considered (S2 Table).

Considering weighted prevalence of severe social isolation according to sex, the decrease was noticeable for males. Non-overlapping confidence intervals in prevalence of severe social isolation in males between ESS 1 and ESS 10 indicated a significant difference, that is, prevalence of severe social isolation in males decreased overtime. Whereas the confidence intervals overlap between ESS 1 and ESS 9 was marginal, i.e., borderline significant. No difference was

**Table 1. Severe social isolation prevalence according to round 1, 9 and 10 of the European Social Survey (ESS).**

| | ESS round (years) | Sex | Population total | Population % | Weighted count | Weighted % (SE) | Weighted 95% C.I. |
|---|---|---|---|---|---|---|---|
| Severe social isolation | 1 (2002–03) | Total | 744 | 2.24 | 626.27 | 2.01 (0.13) | 1.75, 2.27 |
| | | Male | 342 | 2.19 | 274.70 | 1.89 (0.18) | 1.54, 2.24 |
| | | Female | 401 | 2.28 | 351.00 | 2.11 (0.19) | 1.74, 2.49 |
| Severe social isolation | 9 (2018–20) | Total | 715 | 2.03 | 567.56 | 1.77 (0.12) | 1.54, 2.01 |
| | | Male | 260 | 1.56 | 219.57 | 1.38 (0.15) | 1.08, 1.68 |
| | | Female | 455 | 2.46 | 348.00 | 2.16 (0.18) | 1.81, 2.51 |
| Severe social isolation | 10 (2020–22) | Total | 513 | 2.14 | 248.23 | 1.71 (0.14) | 1.44, 1.98 |
| | | Male | 161 | 1.42 | 79.22 | 1.10 (0.15) | 0.81, 1.40 |
| | | Female | 352 | 2.78 | 169.01 | 2.30 (0.22) | 1.88, 2.73 |

*SE*: Standard error, *CI*: Confidence intervals.

observed between ESS 9 and 10 for males and across all ESS rounds for females. Finally, sex differences were found in severe social isolation (i.e., lower in males than females in ESS 9 and 10). Results were almost unchanged when only ten countries with available data in each ESS round were considered (S2 Table).

## Prevalence and trend of severe social isolation by country

Prevalence of severe social isolation in ESS 9 by country is reported in S3 Table. As recommended during the review process, we also examined trends in severe social isolation according to the ten countries with available data for each of the three ESS rounds to further inform the trend analysis (S4 Table). The decrease in total prevalence over time was significant (i.e., confidence intervals do not overlap) for data from Hungary and Slovenia. On the contrary, no other significant trend was found despite overall decreasing (for the Czech Republic, France, Italy, and the Netherlands) and increasing (for Finland, Norway, Portugal, and Switzerland) trends. The data at the country level, especially by sex, should be interpreted with caution considering the low prevalence and wide confidence intervals. The relatively small samples size at the country level (sample size ranges from 919 (Czech Republic) to 1,916 (Netherlands) for ESS 1, from 736 (Portugal) to 1,938 (Italy) for ESS 9, and from 935 (Slovenia) to 1,930 (Czech Republic) for ESS 10) may have decreased statistical power.

## Severe social isolation and sociodemographic characteristics

Sample characteristics and bivariate associations between severe social isolation and the variables of interest are presented in S5 Table.

The first adjusted model included all sociodemographic variables except "years of education" (to avoid redundancy due to the inclusion of the variable "level of education") and "household net income" (due to 19.8% of missing data—approximately 7,000 responses—and availability of similar information from the variable "living uncomfortably on household income").

The results (Table 2) showed that age, living in Central and Eastern Europe, being permanently sick or disabled, doing housework, being retired, born in other country, receiving social benefits/grants as the main source of household income, living uncomfortably on household income, receiving unemployment/redundancy benefit, social benefits/grants, income from

**Table 2. Results of the weighted multivariable regression model of severe social isolation by sociodemographic characteristics.**

| Variable | Estimate (SE) | t-value | Odds ratio (95% C.I.) |
|---|---|---|---|
| Female | 0.36 (0.19) | 1.87 | 1.43 (0.98, 2.09) |
| Age | 0.03 (0.01) | 3.08** | 1.03 (1.01, 1.05) |
| N members household | 0.05 (0.07) | 0.66 | 1.05 (0.91, 1.2) |
| Living alone | 0.19 (0.38) | 0.52 | 1.21 (0.58, 2.55) |
| Living with partner | 0.02 (0.27) | 0.08 | 1.02 (0.6, 1.75) |
| Living with parent | -0.47 (0.36) | -1.3 | 0.62 (0.31, 1.27) |
| Having children | -0.23 (0.27) | -0.85 | 0.79 (0.47, 1.35) |
| Living area | -0.09 (0.08) | -1.12 | 0.91 (0.78, 1.07) |
| *European region* | | | |
| Northern | Reference | | |
| Southern | 0.11 (0.3) | 0.38 | 1.12 (0.62, 2.03) |
| Western | 0.11 (0.24) | 0.47 | 1.12 (0.7, 1.79) |
| Central and Eastern Europe | 0.86 (0.25) | 3.42*** | 2.35 (1.44, 3.84) |
| Level of education | -0.01 (0.06) | -0.15 | 0.99 (0.87, 1.12) |
| Disability | 1.62 (0.27) | 5.93*** | 5.03 (2.95, 8.58) |
| Impairment in daily activities | 0.27 (0.22) | 1.21 | 1.31 (0.85, 2.01) |
| Housework | 0.87 (0.18) | 4.73*** | 2.39 (1.66, 3.42) |
| Retired | 1.53 (0.24) | 6.29*** | 4.6 (2.86, 7.41) |
| Internet use | -0.33 (0.07) | -4.52*** | 0.72 (0.63, 0.83) |
| Born in other country | 0.73 (0.27) | 2.75** | 2.08 (1.23, 3.5) |
| Father/Mother born in other country | -0.14 (0.24) | -0.58 | 0.87 (0.54, 1.4) |
| Father level of education | -0.06 (0.07) | -0.83 | 0.94 (0.81, 1.09) |
| *Father working condition at 14* | | | |
| Employed | Reference | | |
| Unemployed | 0.12 (0.37) | 0.31 | 1.12 (0.54, 2.33) |
| Dead/absent | 0.03 (0.36) | 0.09 | 1.03 (0.51, 2.1) |
| Mother level of education | 0.02 (0.09) | 0.18 | 1.02 (0.85, 1.22) |
| *Mother working condition at 14* | | | |
| Employed | Reference | | |
| Unemployed | 0.04 (0.19) | 0.19 | 1.04 (0.72, 1.5) |
| Dead/absent | -1.01 (0.56) | -1.8 | 0.36 (0.12, 1.09) |
| *Household source of income* | | | |
| Wages, salaries or pensions | Reference | | |
| Unemployment/redundancy benefit | 0.58 (0.37) | 1.58 | 1.79 (0.87, 3.7) |
| Social benefits or grants | 0.95 (0.36) | 2.62** | 2.58 (1.27, 5.27) |
| Income from investments, savings, or other sources | 0.07 (0.44) | 0.17 | 1.08 (0.45, 2.57) |
| Living uncomfortably on household income | 0.46 (0.13) | 3.52*** | 1.58 (1.22, 2.04) |
| *Personal income* | | | |
| Wages, salaries or pensions | Reference | | |
| Unemployment/redundancy benefit | 0.81 (0.38) | 2.15* | 2.25 (1.07, 4.69) |
| Social benefits or grants | 1.17 (0.31) | 3.79*** | 3.23 (1.76, 5.91) |
| Income from investments, savings, or other sources | 1.44 (0.38) | 3.81*** | 4.22 (2.01, 8.85) |
| No income | 1.47 (0.27) | 5.44*** | 4.36 (2.57, 7.42) |

*SE*: Standard error, *C.I.*: Confidence interval.

\* $p < 0.05$

\*\* $p < 0.01$

\*\*\* $p < 0.001$.

investments/savings/other as source of personal income, and having no income were positively associated with severe social isolation. On the contrary, Internet use was negatively associated with severe social isolation. Pseudo-$R^2$ of the model included 0.05 (Cox-Snell), 0.31 (Nagelkerke) and 0.30 (McFadden).

The weighted prevalence of severe social isolation was 2.8% in Central and Eastern Europe whereas it was 0.8% in Northern Europe, 1.5% in Southern Europe and 1.7% in Western Europe. The approximate mean age of the general population was 40 years whereas the mean age of the group in severe social withdrawal was 50 years (S5 Table). Thirteen percent of the general population compared to 18% of the severe social isolation group was born in a country different from that of their residence. Presence of disability (32%), being retired (24%) and doing housework/looking after children/others (32%) were more likely in the severe social isolation group than in the general population (3%, 5%, and 16%, respectively). Twenty-three percent of those in severe social isolation reported social benefits or grants as the main source of household income compared to 3% of the general population. Regarding personal income, of the group with severe social isolation, 36% showed income resulting from wages, salaries or pensions, 8% from unemployment/redundancy benefit, 33% from social benefits or grants, 33% from investments, savings, or other sources, and 15% no income compared to frequencies of 78%, 3%, 5%, 3% and 12%, respectively, among the general population. Finally, the severe social isolation group reported higher worries on present income and lower frequency of internet use compared to the general population.

The multivariable regression model was repeated, as a sensitivity analysis, including household net income. The results that changed include the following: the effects of being born in another country, social benefits/grants as main source of household income and living uncomfortably on household income were no longer significant. Whereas, household net income and number of household members showed negative and positive associations, respectively, with severe social isolation. Values of Pseudo-$R^2$ were almost unaffected.

## Severe social isolation, adverse events and fear

Bivariate associations showed that burglary or assault negatively correlated with severe social isolation whereas feeling unsafe when walking alone positively correlated with severe social isolation (S5 Table). Approximately 17% of the general population and 11% of the group with severe social isolation noted burglary or assault. However, the latter group felt more unsafe when walking alone in their neighbourhood after dark compared to the general population. After controlling for the effect of sociodemographic variables, only feeling unsafe when walking alone was associated with severe social isolation whereas burglary or assault was not significantly associated (Table 3). Pseudo-$R^2$ of the model included 0.05 (Cox-Snell), 0.32 (Nagelkerke) and 0.30 (McFadden).

**Table 3. Results of the weighted multivariable regression model of severe social isolation by adverse events and fear controlling for sociodemographic characteristics (not reported).**

| Variable | Estimate (SE) | t-value | Odds ratio (95% C.I.) |
|---|---|---|---|
| Burglary or assault | -0.48 (0.34) | -1.41 | 0.62 (0.32, 1.21) |
| Feeling unsafe when walking alone | 0.25 (0.11) | 2.36* | 1.29 (1.04, 1.58) |

*SE*: Standard error, *C.I.*: Confidence interval.

* $p < 0.05$

** $p < 0.01$

*** $p < 0.001$.

**Table 4. Results of the weighted multivariable regression model of severe social isolation by emotional support and trust controlling for sociodemographic characteristics (not reported).**

| Variable | Estimate (SE) | t-value | Odds ratio (95% C.I.) |
|---|---|---|---|
| Emotional support | -0.59 (0.07) | -8.26*** | 0.56 (0.48, 0.64) |
| Social trust | -0.21 (0.05) | -4.10*** | 0.81 (0.74, 0.9) |
| Political trust | -0.03 (0.06) | -0.53 | 0.97 (0.86, 1.09) |

*SE*: Standard error, *C.I.*: Confidence interval.

* *p* < 0.05

** *p* < 0.01

*** *p* < 0.001.

## Severe social isolation, emotional support and trust

Participants of the general population reported higher emotional support and political trust than those in severe social isolation (S5 Table). After controlling for the effect of sociodemographic variables, the associations between emotional support, social trust and severe social isolation remained significant whereas the effect of political trust became non-significant (Table 4). Pseudo-$R^2$ of the model included 0.06 (Cox-Snell), 0.38 (Nagelkerke) and 0.36 (McFadden).

## Severe social isolation and well-being

Bivariate associations were found between life satisfaction, happiness, general health, future planning and severe social isolation (S5 Table). Participants with severe social isolation showed lower satisfaction with life and happiness than the general population. On the contrary, they reported higher bad general health and lack of planning about their future compared to those of the general population. After accounting for the effects of sociodemographic characteristics, happiness- negatively- and lack of future planning- positively- were still associated with severe social isolation (Table 5). The associations between severe social isolation and satisfaction with life and bad general health became non-significant. Pseudo-$R^2$ of the model included 0.05 (Cox-Snell), 0.35 (Nagelkerke) and 0.33 (McFadden).

# Discussion

## Prevalence of severe social isolation

The aim of this study was to estimate the prevalence of people in severe social isolation as a proxy for high risk of hikikomori. The use of representative data from the ESS enabled us to

**Table 5. Results of the weighted multivariable regression model of severe social isolation by well-being controlling for sociodemographic characteristics (not reported).**

| Variable | Estimate (SE) | t-value | Odds ratio (95% C.I.) |
|---|---|---|---|
| Satisfaction with life | -0.08 (0.04) | -1.92 | 0.92 (0.85, 1) |
| Happiness | -0.19 (0.05) | -3.49*** | 0.83 (0.74, 0.92) |
| Bad general health | 0.13 (0.13) | 1.05 | 1.14 (0.89, 1.47) |
| Lack of planning for the future | 0.09 (0.03) | 2.67** | 1.09 (1.02, 1.17) |

*SE*: Standard error, *C.I.*: Confidence interval.

* *p* < 0.05

** *p* < 0.01

*** *p* < 0.001.

provide weighted estimates in European countries expanding the current scientific knowledge. We found a prevalence of 1.8% for severe social isolation using weighted data from 29 European countries (ESS round 9) in 2018–20. In the present study, we did not examine hikikomori itself because our analysis relied on data collected from the ESS that did not contain information on marked social isolation at home and its duration. Rather, we investigated severe social isolation as a proxy for hikikomori. It has been previously shown that more than half of participants Not in Employment Education or Training were in a hikikomori condition [16]. Further, we also considered reduced participation in social activities and meetings. The prevalence of 1.8% is in line with that of 1.2–1.9% of hikikomori observed with representative samples in Japan [11,16,26,27] and with that of 1.9–3.2% in China [28,29] and Asian countries overall [38]. Yong et al. [15] showed a prevalence of 6.7% with a working-age (15–64 years) Japanese sample.

Regarding change overtime, the prevalence of severe social isolation was 2% in 2002–03, 1.8% in 2018–20 and 1.7% in 2020–22. Therefore, we provide initial data on the fact that the COVID-19 pandemic might not be affecting severe social isolation, contrary to what was previously hypothesized [39–42]. Severe social isolation was mainly stable overtime between pre-pandemic and pandemic times. Specifically, a decreasing trend was evident even before the pandemic, between 2002–03 and 2018–20. However, future studies are needed to test whether changes in economic stability and inequality could, in turn, influence the prevalence of severe social isolation and hikikomori. Similarly, research with representative samples should confirm these findings at the country level of analysis. Finally, our findings differ from the decline in social connectedness observed in the USA [43] likely because of the different operationalization of isolation (i.e., severe social isolation versus time spent with others).

## Factors associated with severe social isolation

Overtime, prevalence of severe social isolation decreased in males while it remained stable in females. In the ESS round 9 (2018–20), the prevalence was lower among males (1.4%) than females (2.2%). However, sex was no longer associated with severe social isolation after the role of other sociodemographic characteristics was considered. This result is consistent with data reported in previous studies showing no sex difference [15,26] and underscores the role of socio-environmental conditions.

Age and being retired were positively associated with severe social isolation. The association between age and severe social isolation is in line with the result that social isolation affects a significant proportion of the older adult population [44]. However, previous studies including middle and older adults have mostly showed no association between hikikomori and age whereas its association with retirement was not analysed [11,15].

The association between being born in another country and severe social isolation is in line with evidence suggesting poorer social support among migrants [45]. Further, severe social isolation was more likely to occur in participants from Central and Eastern Europe than in individuals from Northern Europe underscoring how the broader socio-cultural country context may exert an impact on individual circumstances [46]. We found that Central and Eastern Europe differed from Northern Europe in the prevalence of three out of five indicators of severe social isolation (S6 Table). Specifically, the prevalence of social meeting with friends, relatives, or colleagues less than once a month or never, not working, and not being in education, was higher in Central and Eastern Europe than in Northern Europe. This point thus to the potential role of socio-economic country-level factors that make individual residents more at risk of severe social isolation as previously hypothesized about loneliness [47–49]. It should be considered that much more previous research analysed differences in loneliness rather than

in social isolation across European countries. And severe social isolation was analysed especially with samples of older adults rather than with the working-age population. Future research may benefit from investigating the potential role of country-level factors such as gross domestic product per capita, social inequality, poverty, social expenditure, and/or education spending (see, for example, the following useful references about the analysis of macro-determinants of NEET and loneliness [50–55]) on the prevalence of severe social isolation. Critically, studies aiming at examining between countries' similarities and differences in risk and protective factors of severe social isolation and hikikomori could be conducted.

Being permanently sick or disabled was associated with severe social isolation. Emerson et al. [56] recently showed that working age English adults with a disability suffered loneliness, low perceived social support and social isolation at significantly higher rates than people without a disability. Longitudinal studies revealed that social isolation and low societal participation predicted future functional disability [57,58]. However, a previous study [15] found no association between sickness and hikikomori.

Housework/looking after children/others was significantly associated with severe social isolation indicating that time spent at home reduces opportunity for social participation and interaction. Social benefits/grants as the main source of household income, living uncomfortably on present income, and personal income resulting from social benefits/grants, income from investment/saving/other sources and having no income were all associated with severe social isolation. These findings are in line with those from previous studies showing associations between no income, unemployment, housework and hikikomori [15,28].

Notably, we found no association between simple indicators of isolation (i.e., number of household members, living alone, living with partner or parents, and having children), poor self-rated health, and severe social isolation, similar to findings of a previous study [15].

It has been shown that that the parents of individuals with hikikomori were more likely to report high education [26]. However, we did not observe such an effect. Bivariate associations indicating lower levels of education of parents of individuals with severe social isolation were no longer significant when accounting for the effect of other sociodemographic variables. Similarly, the bivariate associations between severe social isolation and level of education, shown also by a previous study [29], was no longer significant in the adjusted model.

Of note, the negative association between internet use and severe social isolation suggests that technology access may lower the risk for social isolation [59].

After adjustment for sociodemographic factors, feeling unsafe when walking alone in the neighbourhood after dark was associated with severe social isolation whereas having been the victim of a burglary or assault in the last five years was not associated with severe social isolation. A possible explanation for this finding may be related to the presence of negative beliefs about others/the world, suspiciousness, or paranoid beliefs in individuals with severe social isolation. Indeed, moderate positive associations between loneliness, low social contact and suspiciousness have been demonstrated [60]. At the same time, the above explanation points to the role of social trust and emotional support as possible confounders of the above association, i.e., individuals with severe social isolation could present low social trust and support [61] and this would explain the association between severe social isolation and feeling unsafe when walking alone after dark. Accordingly, we found a consistent association between emotional support and social trust, and severe social isolation. Therefore, to test whether the association between feeling unsafe when walking alone after dark and severe social isolation was still significant after including emotional support and social trust in the model, we ran two post-hoc analyses (not reported in the text). The results showed that the association became non-significant after controlling for social trust, but not for emotional support, supporting the above explanation.

Regarding well-being, severe social isolation was consistently associated with low levels of happiness and planning for the future. These results are consistent with previous studies [62,63] revealing the importance of social participation and interaction for happiness. Furthermore, our findings confirm the importance of external support for future planning probably due to increased self-efficacy and perceived control [64,65]. Accordingly, a recent study [29] demonstrated that hikikomori was negatively associated with positive psychological factors, such as agency and pathways to reach goals, flourishing and positive feelings, and purpose in life. No association was found between severe social isolation and poor general health. This result was likely due to the inclusion of disability and being hampered in daily activities in the adjusted model.

Finally, the ESS survey did not include information about psychopathology. However, the concomitant inclusion of disability ("permanently sick or disable") and being hampered in daily life ("Are you hampered in your daily activities in any way by any longstanding illness, or disability, infirmity or mental health problem?") in the model could provide initial information. Indeed, once considered the effect of disability, the variable being hampered may isolate the effect of "mental health problem". If so, the significant bivariate association between severe social isolation and being hampered in daily life, became non-significant in the adjusted model.

## Implications for hikikomori research

While waiting for representative epidemiological studies using a specific assessment for hikikomori, we believe that the present study constitutes a relevant scientific advancement in knowledge on severe social isolation as a proxy condition for hikikomori.

If avoidance, disinterest or unwillingness, to attend school/work and to participate in social relationships/interactions are not inclusion criteria for hikikomori, individuals with a permanent disability (e.g., physical condition or functional impairment) or doing housework may be at risk of being over-pathologized as hikikomori simply because they spend an increased amount of time at home and may not go out frequently, without these aspects being an indicator of dysfunction (i.e., avoidance, disinterest, or unwillingness, to attend school/work and to participate in social relationships/interactions). In this scenario, we believe that disability and housework should be concomitantly evaluated and considered exclusion criteria for hikikomori. If housework, looking after children or other persons and permanent sickness or disability were considered exclusion criteria of severe social isolation, its overall estimated prevalence, in the present study, would have been 0.7% (95%CI: 0.56, 0.85): 0.63% (95%CI: 0.44, 0.82) among males, and 0.78% (95%CI: 0.56, 0.99) among females. However, considering disability and housework as exclusion criteria of severe social isolation does not seem to be justified in this study since persons with disability or doing housework also suffer the negative consequences of social isolation as examined by reduced social participation and interaction. This issue, clearly, has implications for the definition of hikikomori. Indeed, if avoidance, disinterest or unwillingness, to attend school/work and to participate in social relationships/interactions are inclusion criteria for hikikomori, individuals with a disability or doing housework could still be at increased risk of hikikomori justified by the presence of dysfunction (i.e., avoidance, disinterest or unwillingness, to attend school/work and to participate in social relationships/interactions) with no need to consider disability and housework as exclusion criteria.

The above would also apply to individuals who spent a relevant amount of time at home to help with their children's education, or which are pregnant or giving birth. Indeed, the Cabinet Office of Japan [66] considers the following exclusion criteria of hikikomori: "Individuals whose current state had been triggered by an illness, such as schizophrenia or a physical

disease; those who were pregnant or had recently given birth; those who worked from home; and those who were taking care of their children's education [. . .] Those who stayed home, but who described themselves as a "housewife/husband" or "cleaner" [. . .]" (p.105). Further, should autism spectrum disorder be considered an additional exclusion criterion? If avoidance, disinterest or unwillingness, to attend school/work and to participate in social relationships/interactions are not inclusion criteria for hikikomori, individuals with autism may be at risk of being over-pathologized as hikikomori.

The number of days leaving his/her own home has been proposed to discriminate individuals with and without hikikomori [9]. However, this indicator may not be entirely appropriate in capturing hikikomori, especially after the COVID-19 pandemic, since a person could leave his/her home 2–3 days/week to participate in several social activities, such as part-time working, working sometimes from the office, and/or meeting friends/colleagues for dinner. Could such a person be considered in hikikomori? Is avoidance, disinterest, unwillingness and lack of social participation and interactions the core psychological aspect of hikikomori? Or, is hikikomori not different from social isolation?

To note, avoidance of social situations that provokes marked fear or anxiety is indicative of social anxiety disorder [67]. Further, avoidant behaviours can be extensive (e.g., not going to parties, refusing school) or subtle (e.g., limiting eye contact) and individuals with social anxiety disorder may live at home longer [67]. Efforts have been made to examine whether avoidant personality disorder- characterized by social inhibition, feelings of inadequacy and hypersensitivity to negative evaluation- is a clinically useful diagnosis, distinct from social phobia [68,69]. According to the severity continuum hypothesis, the two disorders would differ only in severity [70–74]. However, a qualitative [75–77] distinction has been proposed in addition to a quantitative one [78–80]. Overall, previous studies support a strong relationship between social anxiety and avoidant personality disorder, with the latter being probably a severe variant of the former. Could this also apply to hikikomori? That is, could hikikomori be a severe variant of social anxiety disorder [81]? Interestingly, Cox et al. [82] found that mood disorders were particularly common in individuals with comorbid social anxiety and avoidant personality disorder. Further, a previous relevant study conducted with a clinical sample of Spanish adults who endorsed criteria for hikikomori found that all participants showed a comorbid condition, mainly psychotic, personality, affective and anxiety disorders [24].

As pointed out above, avoidance, disinterest or unwillingness to attend school/work and to participate in social relationships/interactions, and absence of other primary mental disorders have been considered core criteria of hikikomori [11–13,66]. Nevertheless, the most recent proposal of a diagnostic criteria of hikikomori [83] reports that: "The requirement for avoidance of social situations and relationships has been removed. In our interviews assessing individuals for hikikomori, they commonly report having few meaningful social relationships and little social interaction, but deny avoiding social interaction. Many clinicians often wonder about what distinguishes hikikomori from social anxiety disorder, and this lack of avoidance is one of the primary differences." (p.117). Unfortunately, no evidence has been published so far indicating low avoidance of social participation and interactions in hikikomori and some of the same authors have confirmed, in a subsequent article, that "individuals with hikikomori avoid social situations voluntarily" [39] (p.506). It could be the case that individuals who will develop hikikomori are conscious of avoiding social situations only at the beginning of the symptomatology and that, once the hikikomori condition is stabilized, they do not feel the urge to avoid social situations because they are no longer participating in them. This aspect needs to be addressed by future research.

Some studies found evidence on the role of avoidant personality traits/disorder in hikikomori [25,84]. Most importantly, individuals with social anxiety disorder may endured social

situations with intense fear or anxiety and not inevitably avoid them [67]. Thus, it is not clear in which way the exclusion of the above criterion illuminates the difference between social anxiety disorder and hikikomori. On the contrary, it leads to viewing hikikomori as equal to a severe form of social isolation.

Further, stressful or humiliating experiences (e.g., bullying) may exert a prominent role in the onset of both social anxiety and avoidant personality disorder [67]. Similar findings were observed in hikikomori [28,33]. Accordingly, the possibility that adjustment disorder could take the form of social withdrawal and mimic a hikikomori condition should also be taken into account [85].

Despite the fact that a detailed discussion of the relationship between hikikomori and psychopathology is beyond the scope of this article, Fig 1 may be of help in clarifying how hikikomori may be a combination of social isolation (behavioural-objective symptom) and social withdrawal (psychological-subjective symptom) and motivating future research on hikikomori criteria [10]. Relevant efforts are needed to clarify similarities and distinctions between hikikomori and other forms of psychopathology [86]. Similarly, another issue regards whether NEET individuals with hikikomori (social isolation and withdrawal) constitute a NEET subgroup most in need of support. Future studies should explore the sociodemographic and clinical characteristics of this specific NEET subgroup compared to that of NEET individuals who maintain social interactions and are socially active.

## Study limitations

Some limitations of this study need to be considered when interpreting the above results. First, this is a secondary analysis of data collected for study aims different than studying severe social isolation. Consequently, questions were not formulated in the way most favourable for the

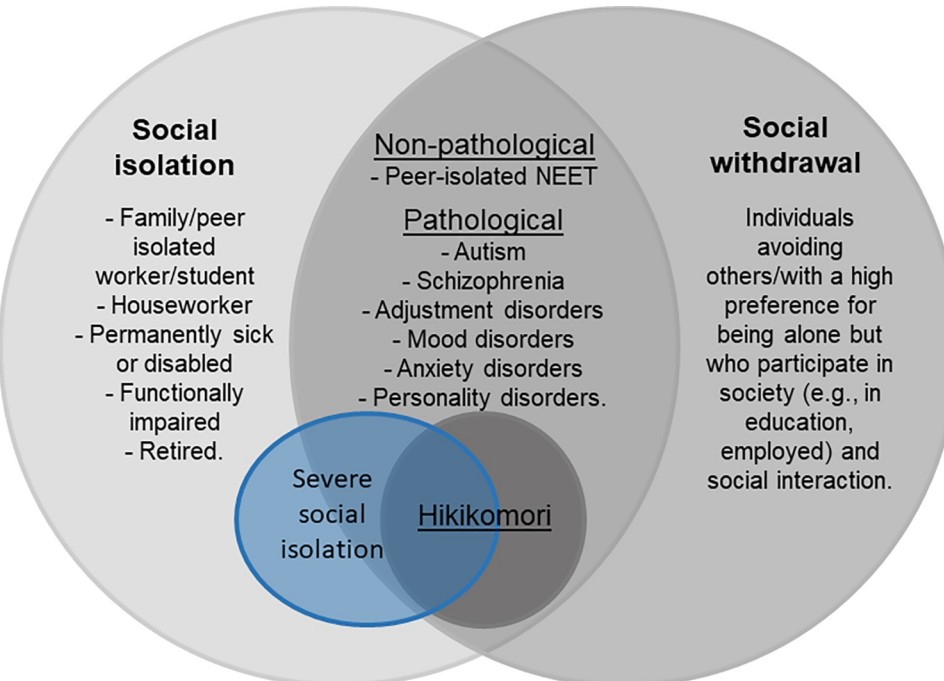

**Fig 1. Hikikomori as a combination of social withdrawal and social isolation.** *Note*: Dimension does not represent real proportion. In each circle, expected membership categories are reported. Nevertheless, the distinction is not intended to be rigid. Severe social isolation, as operationalized in the present study, may include individuals belonging to different categories (in blue).

study of severe social isolation, and no standardized questionnaire for its data collection was used. Second, as reported above in the text, we investigated severe social isolation as a proxy for hikikomori. It was possible because we also considered reduced social participation (i.e., being not in education and unemployed) and interaction (i.e., reduced social meetings and activities) in defining severe social isolation. However, no specific questionnaire or clinical diagnostic interview investigated hikikomori symptoms or criteria. Further, significant impairment or distress caused by social isolation was not included in the definition of severe social isolation in line with the fact that we examined a condition at high risk for hikikomori rather than hikikomori itself. It should be noted that most studies (88.5%) examining hikikomori, in reality, did not consider significant impairment or distress [14] and, thus, the validity of their findings is questionable. The inclusion of specific psychological indicators of dysfunction and impairment/distress caused by social withdrawal and isolation is needed in future studies of hikikomori [10,14]. Third, our result on the prevalence of severe social isolation for the young age group of 15–29 years (0.36%, 95%CI: 0.2, 0.52) may represent an underestimation of the phenomenon in this age group since there may be youths who are formally in education but in practice do not attend classes and do not take part in social activities. Indeed, if the age group 15–19 years was considered, the prevalence slightly decreased (0.27%, 95%CI: 0.00, 0.54). Consequently, adopting a conservative approach (i.e., including lack of social interaction and participation for severe social isolation), we were not able to detect this specific subgroup of young people in severe social isolation. However, our prevalence of severe social isolation in adolescents and young adults is in line with a prevalence of 0.2% of students (academic year 2020–2021) who received a certificate of social withdrawal from local mental health departments in Italy according to a recent report [87] and higher than the approximate estimate of 0.025% based on Emilia-Romagna students (academic year 2017–2018) who rarely leave their home and do not attend school according to a previous report [88,89]. There is yet confusion on the definition of hikikomori because different definitions have been proposed [10]. Unfortunately, this may erroneously lead to consider as hikikomori those adolescents who attend school [87,90]. If it is true that adolescent who leave their home only to attend school may have vulnerabilities or be at risk for social and mental health problems, during a post-pandemic period it could be better to err on the side of caution avoiding over pathologizing stressful reactions of young people and/or conditions not clearly defined [86,88]. Finally, the sample is composed of persons residing within private households and this constitute an additional limitation of the present study. It is thus plausible that this sample does not represent people at greater risk for severe social isolation (e.g., patients from psychiatric/therapeutic communities, homeless people, asylum seekers, refugees and migrants).

## Supporting information

**S1 Table. List of indicators used in this study.**
(DOCX)

**S2 Table. Severe social isolation prevalence according to rounds 1, 9 and 10 of the European Social Survey (ESS) for ten countries with available data in each survey (Czech Republic, Finland, France, Hungary, Italy, Netherlands, Norway, Portugal, Slovenia, Switzerland).** SE: Standard error, CI: Confidence intervals.
(DOCX)

**S3 Table. Severe social isolation prevalence according to country (European Social Survey round 9).** CI: Confidence intervals.
(DOCX)

**S4 Table. Severe social isolation prevalence according to rounds 1, 9 and 10 of the European Social Survey (ESS) for each of the ten countries with available data (Czech Republic, Finland, France, Hungary, Italy, Netherlands, Norway, Portugal, Slovenia, Switzerland).** CI: Confidence intervals.
(DOCX)

**S5 Table. Weighted descriptive statistics of sample characteristics and tests for differences' results according to the absence/presence of severe social isolation.** df: Degree of freedom, M: Mean, SE: Standard error. * p< 0.05, ** p< 0.01, *** p< 0.001.
(DOCX)

**S6 Table. Weighted descriptive statistics of indicators of severe social isolation according to the European region.** df: Degree of freedom. * p< 0.05, ** p< 0.01, *** p< 0.001.
(DOCX)

## Acknowledgments

The authors would like to thank the European Social Survey for publicly sharing the data used in this study at https://ess-search.nsd.no/.

## Author Contributions

**Conceptualization:** Simone Amendola.

**Data curation:** Simone Amendola.

**Formal analysis:** Simone Amendola.

**Methodology:** Simone Amendola.

**Project administration:** Simone Amendola.

**Supervision:** Rita Cerutti, Agnes von Wyl.

**Visualization:** Rita Cerutti, Agnes von Wyl.

**Writing – original draft:** Simone Amendola.

**Writing – review & editing:** Simone Amendola, Rita Cerutti, Agnes von Wyl.

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
