## [Decision Letter · Decision Letter 0]

21 Jun 2023

PONE-D-23-11201Estimating the prevalence and characteristics of people in severe social isolation in 29 European countries: A secondary analysis of data from the European Social Survey round 9 (2018-2020)PLOS ONE

Dear Dr. Amendola,

Thank you for submitting your manuscript to PLOS ONE. After careful consideration, we feel that it has merit but does not fully meet PLOS ONE’s publication criteria as it currently stands. Therefore, we invite you to submit a revised version of the manuscript that addresses the points raised during the review process.

The major revision of the paper is required. In addition to addressing reviewer #2 comments (see below for details), authors need to address fully the concerns raised by reviewer #1 in order for paper to be considered for publication. Namely, authors need to justify the conceptualization, validity, added value and measurement of the concept hikikomori in European context in a convincing manner grounded on the previous research. Specifically, there is a need to clarify the difference of the concept from loneliness and extreme social isolation.Furthermore, authors need to offer a more broad socio-cultural explanation of the difference in its prevalence across countries in Europe.

We look forward to receiving your revised manuscript.

Kind regards,

Srebrenka Letina, Ph.D.

Academic Editor

PLOS ONE

2. We noted in your submission details that a portion of your manuscript may have been presented or published elsewhere. [However, as reported in the title, abstract and main document, this is a secondary analysis of data from the European Social Survey.] Please clarify whether this publication was peer-reviewed and formally published. If this work was previously peer-reviewed and published, in the cover letter please provide the reason that this work does not constitute dual publication and should be included in the current manuscript.

Additional Editor Comments:

Major revision required.

Reviewers' comments:

Reviewer's Responses to Questions

**Comments to the Author**

1. Is the manuscript technically sound, and do the data support the conclusions?

Reviewer #1: No

Reviewer #2: Yes

2. Has the statistical analysis been performed appropriately and rigorously? 

Reviewer #1: Yes

Reviewer #2: Yes

3. Have the authors made all data underlying the findings in their manuscript fully available?

Reviewer #1: Yes

Reviewer #2: Yes

4. Is the manuscript presented in an intelligible fashion and written in standard English?

Reviewer #1: Yes

Reviewer #2: Yes

5. Review Comments to the Author

Reviewer #1: The article’s main aim is to estimate high social isolation in Europe and posit its relevance for hikikomori. In addition, it delves into the factors that predict high social isolation.

Operationalization: the definition of hikikomori includes also the subjective feeling of isolation (e.g. the feeling of ”distress”, but this subjective dimension is not included in the authors’ measure of extreme isolation. The reason for and implications of leaving out this ”loneliness” or other type of mental subjective consequences of social isolation need to be discussed, since this is an essential feature of hikikomori. Also the indicators used to measure social isolation here (few social meetings, few social activities, not working, not job seeking, not in education); the readers do not see how these are put together into an index. Is it an additive index or constructed from factors? Or is it a multiplicative index, because hikimori symptoms should be found in combination? This is crucial to understand the claim. With this information (and the subjective dimension) missing, I find it hard to go along with the central claim, that this measure of extreme social isolation is a proxy for hikikomori. This paper is actually about extreme social isolation in Europe. I find the hikikomori aspect of this both distracting, not supported, and perhaps less theoretically interesting than the more general issue of extreme social isolation. What additionally is gained by the ”hikikomori” concept in a european context?

Theory:

If this were an example of hikikomori, if this operationalization were convincing, what can we do with these findings? We learn that hikomori is also present in Europe, we learn about some of determinants, but we do not know why, theoretically, its prevalence varies. For example, there is the evidence that this isolation is more widespread in Central and Eastern Europe, but we do not get a discussion about why. There is growing social-scientific research about why social isolation and loneliness vary culturally, but none of this work is cited. As such, we get a list of independent variables and their impacts, but readers are left to guess why they are included and how to interpret their findings. My advice would be to make use of sociological, cultural, and comparative social-psychological findings on these themes to try to disentangle where enhanced social isolation is found in Europe and why. Likewise, I was not convinced by the brief trend analysis, going against a growing vast research, that suggests social isolation did not increase during COVID years. An analysis of this type would really have to take subgroups into account, rather than country-level averages, to look at social isolation changes within subgroups.

Reviewer #2: Thank you for the opportunity to review this manuscript “Estimating the prevalence and characteristics of people in severe social isolation in 29 European countries: A secondary analysis of data from the European Social Survey round 9 (2018-2020)”. This study thoughtfully estimates the prevalence of people in severe social isolation as a proxy for high risk of hikikomori using data from 29 European countries. Additionally, it considers the impacts of relevant psychological and social variables on social isolation.

Line 49-51. Social isolation is defined and references as a objective and subjective construct. Consider revising this text as much of current think considers social isolation to be an objective state and strives to distinguish this from loneliness. This is described in the recent NASEM report on social isolation and loneliness as well as Age UK information- https://www.ageuk.org.uk/our-impact/policy-research/loneliness-research-and-resources/loneliness-isolation-understanding-the-difference-why-it-matters/.

Line 164- 166. Authors have opportunity to be more specific that this is referring to emotional support rather than broader domain of social support based on the question that is included in the study.

Line 343-344. Authors make a loneliness reference however this conceptually differs from social isolation.

6. PLOS authors have the option to publish the peer review history of their article (what does this mean?). If published, this will include your full peer review and any attached files.

Reviewer #1: No

Reviewer #2: No

---

## [Author Response · Author response to Decision Letter 0]

26 Jul 2023

Please refer to the files "Cover letter_R1" and "Response to Reviewers".

---

## [Decision Letter · Decision Letter 1]

29 Aug 2023

Estimating the prevalence and characteristics of people in severe social isolation in 29 European countries: A secondary analysis of data from the European Social Survey round 9 (2018-2020)

PONE-D-23-11201R1

Dear Dr. Amendola,

We’re pleased to inform you that your manuscript has been judged scientifically suitable for publication and will be formally accepted for publication once it meets all outstanding technical requirements.

Kind regards,

Srebrenka Letina, Ph.D.

Academic Editor

PLOS ONE

Additional Editor Comments (optional):

Reviewers' comments:

Reviewer's Responses to Questions

**Comments to the Author**

1. If the authors have adequately addressed your comments raised in a previous round of review and you feel that this manuscript is now acceptable for publication, you may indicate that here to bypass the “Comments to the Author” section, enter your conflict of interest statement in the “Confidential to Editor” section, and submit your "Accept" recommendation.

Reviewer #1: All comments have been addressed

Reviewer #2: All comments have been addressed

2. Is the manuscript technically sound, and do the data support the conclusions?

Reviewer #1: Yes

Reviewer #2: Yes

3. Has the statistical analysis been performed appropriately and rigorously? 

Reviewer #1: Yes

Reviewer #2: Yes

4. Have the authors made all data underlying the findings in their manuscript fully available?

Reviewer #1: Yes

Reviewer #2: Yes

5. Is the manuscript presented in an intelligible fashion and written in standard English?

Reviewer #1: Yes

Reviewer #2: Yes

6. Review Comments to the Author

Reviewer #1: The author(s) have adequately addressed all of my concerns, including more nuanced discussion and analysis of points that I found problematic. In particular, shifting the framing to (severe) social isolation as a risk factor for hikimori (rather than hikimori itself) solved some important concerns.

Reviewer #2: (No Response)

7. PLOS authors have the option to publish the peer review history of their article (what does this mean?). If published, this will include your full peer review and any attached files.

Reviewer #1: No

Reviewer #2: No

---

## [Editor Report · Acceptance letter]

4 Sep 2023

PONE-D-23-11201R1 

Estimating the prevalence and characteristics of people in severe social isolation in 29 European countries: A secondary analysis of data from the European Social Survey round 9 (2018-2020) 

Dear Dr. Amendola:

I'm pleased to inform you that your manuscript has been deemed suitable for publication in PLOS ONE. Congratulations! Your manuscript is now with our production department. 

Kind regards, 

on behalf of

Dr. Srebrenka Letina 

Academic Editor

PLOS ONE